# Exendin-4 enhances GLP-1 signaling and reduces anxiety-like behaviors in male heroin withdrawal mice

Yang Xiang◉, Xiaowei Yan◉, Rongrong Li, Chunlu Li, Dan Yin, Cancan Luo, Yingqi Xiong, Yan Hong*, Yixin Li*, Baijuan Xia◉*

Department of Histology and Embryology, School of Basic Medical Sciences, Guizhou Medical University, Gui'an New District, Guizhou, China

◉ These authors contributed equally to this work.
* xiabaijuan4191@163.com (BX); hongyanb@163.com (YH); liyixin@gmc.ed.cn (YL)

## Abstract

Anxiety and depression significantly contribute to heroin relapse, and addressing these issues could lower relapse rates. The basolateral amygdala (BLA) and nucleus tractus solitarius (NTS) are involved in regulating these emotions, but the molecular mechanisms during heroin withdrawal are not yet understood. Subcutaneous injection of heroin into C57BL/6J mice to simulate chronic dependence, withdrawal, and Exendin-4 treatment. Assess anxiety and depression-like behaviors using open field test (OFT), elevated plus maze (EPM), forced swimming test (FST), and tail suspension test (TST). Analyze neuronal and protein expression changes in the BLA brain area with Western blotting (WB) and immunofluorescence staining. Heroin dependence reduces glutamatergic neurons in BLA without affecting anxiety and depression-like behaviors, due to the inhibitory effect of heroin reward. During withdrawal, GLP-1 secretion by the NTS rises, increasing c-Fos and GLP-1 receptor expression in glutamatergic neurons of BLA, linked to heightened anxiety but not depression. A 7-day treatment with Exendin-4 (2 μg/kg) alleviates anxiety in withdrawal mice by downregulating GLP-1 signaling in the NTS-BLA circuit, indicating GLP-1's role in regulating anxiety during heroin withdrawal. GLP-1 receptors within BLA may serve as molecular targets for modulating emotional states, thereby offering empirical support for strategies aimed at preventing heroin relapse.

## 1. Introduction

The persistent and stable prevalence of heroin dependence, a significant opioid drug, in recent years may be attributed to the notably high relapse rates observed among individuals with heroin use disorder [1]. Research indicates that following heroin detoxification treatment, the relapse rate can reach up to 59% within a five-year period [2], potentially linked to the negative emotions experienced during

**Data availability statement:** All relevant data are within the paper and its Supporting information files.

**Funding:** This work was primarily supported by the Guizhou Provincial Science and Technology Program Project (Grant No. QianKeHeJiChu ZK (2024) General 123) awarded to BX, National Natural Science Foundation Cultivation Project (Grant No. 21NSFCP37) awarded to BX and National Natural Science Foundation of China grants (Grant Nos.82201653) awarded to RL, which provided direct funding for the experiments reported in this manuscript. In addition, several National Natural Science Foundation of China grants (Grant Nos. 81960253 and 82160268) awarded to BX and YL and Key Laboratory for Research on Autoimmune Diseases of Higher Education Schools in Guizhou Province, (Qianjiaoji [2023] 016) awarded to BX were held by members of our research group during the study period. These grants offered general support to the laboratory but did not directly fund the specific research activities presented here. The funders had no role in study design, data collection and analysis, decision to publish, or preparation of the manuscript.

**Competing interests:** The authors have declared that no competing interests exist.

heroin withdrawal. Among these negative emotions, anxiety and depressive moods are prominent components following opioid withdrawal [3,4]. According to Dan Luo, among patients with an average of eight years of chronic heroin dependence, 57.9% experience anxiety and 69% experience depression [5]. This phenomenon is also observed in animal disease models. For instance, Dersu Ozdemir and colleagues reported that natural withdrawal from opioids, such as morphine and fentanyl, can induce significant anxiety and depression-like behaviors [3,4]. However, in murine models, the anxiety and depressive emotions and their underlying mechanisms following natural heroin withdrawal require further investigation. Given that the comorbidity rate of anxiety disorders and depression is as high as 60% [6], it is appropriate to study these conditions concurrently. We propose that effective interventions targeting anxiety and depression could help lower relapse rates among opioid users, including heroin addicts [3,7,8].

The basolateral amygdala (BLA), a critical component of the amygdala within the brain's limbic system, plays a central role in emotion processing, memory formation, and fear conditioning [9,10]. Previous research has indicated that projections from the ventral tegmental area (VTA) to the BLA are implicated in the regulation of anxiety-like behaviors [11], and that inflammation within the BLA can intensify anxiety and depressive behaviors in mice [12]. Furthermore, existing studies have shown that opioid dependence can influence the epigenetic landscape, morphological structure, and neurotransmitter secretion in brain regions such as the prefrontal cortex (PFC), Nucleus Accumbens (NAc), and amygdala [8,13,14]. Nonetheless, the effects of chronic heroin dependence and its subsequent withdrawal on the BLA region in mice remain to be thoroughly explored. The NTS, situated in the brainstem, serves as a pivotal center for the internal autonomic nervous system and the regulation of emotions [15,16]. The connectivity between the NTS and emotion-related regions, such as the amygdala, facilitates the processing and transmission of emotional signals, thereby modulating anxiety [15]. According to research by Li, Shaoyuan, the activation of sensory neurons in the NTS affects regions such as the amygdala via the corpus callosum, resulting in a reduction of depressive symptoms [16]. Nevertheless, the specific role of the NTS in the context of heroin withdrawal remains inadequately understood [17,18]. Prior studies have demonstrated that neural circuits such as the BLA-PFC, VTA-BLA, and hippocampus(Hip)-BLA are involved in the regulation of anxiety and depressive states [6,10,19]. However, there is a paucity of research on the role of the NTS-BLA circuit in regulating anxiety and depression, particularly in the context of heroin withdrawal in mice, indicating a need for further investigation.

Glucagon-like peptide-1 (GLP-1) is a prominent gut-brain peptide predominantly secreted by preproglucagon (PPG) neurons within the NTS of the central nervous system (CNS) [20]. Within the CNS, GLP-1 is known for its anti-inflammatory, antioxidant, neuroprotective, and neurogenic properties [21–23]. Furthermore, it has been demonstrated to exert anxiolytic and antidepressant effects [24,25]. However, administration of GLP-1 at elevated doses may exacerbate anxiety and depressive behaviors in a dose-dependent manner [26–28]. These effects are mediated through the activation of its receptor, GLP-1R, which is extensively distributed across

various brain regions, including the NAc, BLA, PFC, and VTA [20]. Recent studies have shown that GLP-1 analogs, such as Exendin-4, effectively reduce the self-administration of heroin, alcohol, and cocaine in rodent models of addiction [29–33]. Nonetheless, there is a paucity of research regarding the effects of heroin withdrawal on GLP-1 secretion in the murine brain. Further investigation is warranted to ascertain whether Exendin-4 intervention can mitigate anxiety and depression-like behaviors in mice undergoing heroin withdrawal.

This study explored the role of the GLP-1 signaling pathway from NTS to BLA and the impact of Exendin-4 on anxiety and depression-like behaviors in mice undergoing heroin withdrawal. By establishing mouse models of heroin dependence and withdrawal, researchers used behavioral tests, immunofluorescence, and WB to assess changes in symptoms, anxiety, and protein expression. Findings revealed heroin dependence reduces glutamatergic neurons in the BLA, potentially triggering neural repair during withdrawal. Withdrawal increased anxiety-like behaviors, possibly linked to the upregulation of the GLP-1 pathway. Exendin-4 treatment for 7 days reduced anxiety-like behaviors, associated with increased glutamatergic neurons and downregulated GLP-1 signaling in NTS-BLA. This suggests a link between GLP-1 signaling and anxiety during heroin withdrawal.

## 2. Methods

### 2.1. Animals

In this study, we used male C57BL/6J mice, 8 weeks old and weighing around 22g, sourced from SPF (Beijing) Biotechnology Co., Ltd. We selected mice with similar fur, walking, and social conditions for the experiment. They were kept in a 12-hour light-dark cycle (lights on at 8:00 am and lights off at 8:00 pm) with free access to sterilized food and water, grouped together throughout the experiment. After the behavioral experiment, mice were euthanized by intraperitoneal injection of 1.25% Avertin (0.2 ml/10 g) and head and neck dislocation, and then samples were taken for the experiment. All surgeries were performed under anesthesia with 1.25% Avertin, and every effort was made to minimize pain in the mice. All animal procedures were conducted in accordance with the "Guidelines for Ethical Review of Animal Welfare" (GB/T 35892−2018, China) and approved by the Experimental Animal Ethics Committee of Guizhou Medical University (No.2200040).

### 2.2. Reagents

Heroin (purity 80.5%) was acquired from the Guizhou Public Security Bureau, dissolved in sterile saline to prepare a stock solution of 10 mg/ml, and stored at 4°C. Exendin-4 (HY-13443) was purchased from MCE, dissolved in sterile saline to prepare a stock solution of 10 mg/ml, and stored at −20°C. Super ECL Plus ultra-sensitive luminescent solution (P1050-100) was purchased from APPLYGEN.

Antibodies. c-Fos (T62069M) was obtained from Abmart; GLP-1 (BF8043) and CREB (BF8028) were sourced from Affinity; GLP-1R (IPB3238) was acquired from Baijia; TrkB (S-488-5) was obtained from STARTER; GAD1 (67648–1-lg) and VGLUT1 (55491–1-AP) were purchased from Proteintech; BDNF (GTX132621) were sourced from GeneTex; Goat anti-rabbit IgG-HRP (BS13278) and Goat anti-mouse IgG-HRP (BS12478) were obtained from Bioworld; Alexa Fluor 488-conjugated Goat anti-mouse (H + L) (A0428) and Cy3-conjugated Goat anti-rabbit IgG (H + L) (A0516) were sourced from Beyotime; Dylight 488-conjugated AffiniPure Goat anti-rabbit IgG (H + L) was purchased from BOSTER; cy3-conjugated Goat anti-mouse IgG polyclonal antibody (HA1109) was obtained from Hua An.

### 2.3. Virus

The AAV virus used in this study was pAAV-hSyn-EGFP-2A-Cre-WPRE, purchased from Vigene Biosciences (Shanghai) Co., Ltd. A total of 0.2 μl of pAAV-hSyn-EGFP-2A-Cre-WPRE virus or physiological saline was injected into the BLA brain region at a rate of 0.1 μl/5 min. The injection coordinates for the BLA were as follows: Anteroposterior (AP): −1.46

mm (Bregma); Mediolateral (ML): 2.7 mm; Dorsoventral (DV): 4.5 mm. Virus-mediated protein expression was detectable under a microscope after three weeks, reaching peak expression at this time and lasting for over three months. Subsequently, tissue was frozen and sectioned, and the injection site was examined under a high-resolution confocal microscope [6].

### 2.4. Test of withdrawal symptoms in heroin withdrawal mice

Sixteen male C57BL/6J mice were randomly assigned to either a heroin natural withdrawal group (H, n = 8) or a saline control group (S, n = 8). Heroin dependence was induced over a period of 10 days. Twenty-four hours following the cessation of heroin administration, the mice were placed individually in a transparent plastic container with a diameter of 20 cm and a height of 30 cm. Their activities were recorded for a duration of 10 minutes using a camera [34]. During this observation period, the number of behaviors associated with withdrawal, including standing positions, licking hair, face washing and bowel movements, was documented and subsequently analyzed.

### 2.5. Mouse heroin dependence model and behavioral testing

Sixteen 8-week-old male C57BL/6J mice with similar body weights were randomly assigned to either a heroin-dependent group (H', n = 8) or a normal saline control group (S', n = 8). The groups were compared for body weight, fur condition, and walking status to confirm successful randomization. Heroin dependence was induced using Bailey et al.'s method [35], with heroin injections given subcutaneously in increasing doses over ten days. The dosage schedule was 2 × 1 mg/kg on day 1, 2 × 2 mg/kg on days 2–3, 2 × 4 mg/kg on days 4–5, and 2 × 8 mg/kg twice daily on days 6–10. The control group received saline injections. To evaluate chronic heroin dependence, anxiety and depression tests were conducted on days 9 and 10. The open field test took place on the morning of day 9, followed by the elevated plus maze test in the afternoon. The more intense forced swim and tail suspension tests were conducted on day 10. Mice were given at least 2 hours of rest after each test to reduce the influence of prior experiments.

### 2.6. Selection of Exendin-4 concentration

To investigate the neural repair effects of Exendin-4 without influencing blood glucose levels and body weight in mice, the study was conducted as follows: (1) Twelve mice were randomly assigned to two groups (n = 6), each receiving different doses of Exendin-4 (2 µg/kg and 4 µg/kg). (2) Based on the blood glucose results, the mice were subsequently re-randomized into a normal saline control group and a 2 µg/kg Exendin-4 intervention group (n = 6). The intervention was administered in the morning and evening, and the mice were weighed every two days. Statistical analysis was performed on the weight changes between the two groups.

### 2.7. Heroin withdrawal model, Exendin-4 intervention, and behavioral testing

Forty-eight male C57BL/6J mice were randomly divided into a heroin-dependent group (n = 24) and a saline control group (n = 24). Initial assessments of body weight, fur condition, and locomotor activity confirmed successful randomization. The heroin-dependent group then received chronic heroin exposure for 10 days. Based on the methodology outlined by Ozdemir, Jerome A., and colleagues [3,4], a natural withdrawal period of 7 days was implemented without any additional interventions, apart from collective feeding. The control and experimental groups were divided into cohorts of 8 mice each for natural withdrawal over 7 days, followed by behavioral tests. Remaining mice were split into two subgroups for a 7-day Exendin-4 intervention, with the control group receiving saline. This created four conditions: Sal + Sal, Sal + Ex-4, Her + Sal, and Her + Ex-4. Post-intervention, behavioral tests were conducted in this order: open field, elevated plus maze, tail suspension, and forced swimming, with one test per day between 10 a.m. and 4 p.m. to avoid prior test influence.

## 2.8. Behavioral tests

**2.8.1. Elevated plus maze test (EPM).** The apparatus consists of a white acrylic board with two open arms (dimensions: 25 cm × 5 cm × 0.5 cm), two closed arms (dimensions: 25 cm × 5 cm × 16 cm), and a central open platform (dimensions: 5 cm × 5 cm). Prior to the commencement of behavioral testing, all experimental mice were acclimated to the testing environment by being transferred to the behavioral testing room 30 minutes beforehand, during which a white noise generator was activated to facilitate adaptation. The testing is conducted in a soundproof room illuminated by white light. The elevated plus maze (EPM) apparatus is meticulously cleaned with an odorless 50% ethanol solution. Each mouse is placed at the center of the maze, and video tracking is performed for a duration of five minutes. The time spent, movement, and distance traveled in the open arms are quantified using the Tracking Master V5.3.7 video tracking system [36].

**2.8.2. Open field test (OFT).** At the onset of behavioral testing, all experimental mice were acclimatized to the behavioral testing room for 30 minutes prior to the commencement of the experiments, during which a white noise generator was activated to facilitate adaptation to the testing environment. The mice were subsequently placed in an open field apparatus measuring 50 cm × 50 cm for a duration of 5 minutes, during which various parameters were assessed, including walking distance, time spent, and frequency of entries into the central area, which measures 25 cm × 25 cm, across different groups of mice. The equipment was meticulously sanitized by hand using an odorless 50% ethanol solution. Data collection was conducted using the Tracking Master V5.3.7 video tracking system, which monitored and recorded the distance traveled, frequency of entries, and duration spent by the mice in the center of the open field [37].

**2.8.3. Forced swimming test (FST).** At the onset of behavioral testing, all experimental mice were relocated to the behavioral testing room 30 minutes prior to the commencement of the experiment to acclimate to the environment, facilitated by the activation of a white noise generator. A white background plate was selected, and a transparent cylindrical container with dimensions of 30 cm in height and 20 cm in diameter was positioned accordingly. The container was filled with water maintained at a temperature of 23–25°C to a depth of 20 cm. Each mouse was individually placed into a separate container, allowed to acclimate for 2 minutes, and subsequently recorded for a duration of 4 minutes using a camera. Upon completion of each trial, the water was replaced before proceeding with the subsequent test. The stationary time within the 4-minute recording period was documented and analyzed using Tracking Master V5.3.7 software [38].

**2.8.4. Tail suspension test (TST).** At the beginning of behavioral testing, all experimental mice were transferred to the behavioral testing room 30 minutes in advance and the white noise generator was turned on to adapt to the conditions of the behavioral testing room. Select a white background board, place the suspended tail test stand in front, hang the animal's tail with tape, wait for 2 minutes, and then record with the camera for 4 minutes. After the experiment, clean the environment and spray 50% alcohol to remove odor. Then, record the next mouse test video and use Tracking Master V5.3.7 software to analyze the last 4 minutes of rest time [39].

## 2.9. Immunofluorescence staining

Mice were anesthetized with 1.25% Avertin (0.2 ml/10 g) and perfused with saline and 4% paraformaldehyde. Brain tissues were collected, fixed in 4% paraformaldehyde for at least 24 hours, dehydrated in graded alcohol, cleared with xylene, and embedded in paraffin. Paraffin blocks were trimmed and sectioned at 6 μm thickness. Sections were mounted and baked at 38°C for 48 hours. The sections were immersed twice in xylene and absolute ethanol for 10 minutes each, then in 95%, 85%, and 75% ethanol for 5 minutes each, followed by a 15-minute PBS rinse. A 5% goat serum solution was applied for 30 minutes at room temperature. After removing the blocking solution, primary antibodies (GLP-1R and c-Fos at 1:100; VGLUT1 and GAD1 at 1:200) were added and incubated overnight at 4°C. The next day, sections were washed with PBS and incubated with fluorescent secondary antibodies (488/CY3 at 1:500) for 120 minutes at room temperature, then wash three times with PBS, each time for 15 minutes. Sections were dried, mounted with anti-fade medium

containing DAPI, and secured with nail polish. Finally, they were examined using a high-resolution spinning disk confocal microscope, and fluorescence intensity and cell count were analyzed with ImageJ.

## 2.10. Western blotting

Mice were anesthetized with 1.25% Avertin, and their whole brains were chilled on ice. The BLA and NTS regions were dissected for WB experiments. Proteins were extracted with a lysis buffer, quantified using the BCA assay, and standardized to 30 µg per sample. This was repeated four times, ensuring internal reference proteins like GAPDH were consistent across samples before detecting target proteins. Proteins were then separated by SDS-PAGE and transferred to a PVDF membrane. The membrane was blocked with 5% non-fat milk for 2 hours, then incubated overnight with primary antibodies (GLP-1R, GLP-1, BDNF, TrkB, CREB at 1:1000 dilution; GAPDH at 1:2500). It was then treated with HRP-conjugated secondary antibodies (1:2500 in TBST) for 1.5 hours at room temperature. Immunoblots were visualized with enhanced chemiluminescence, and gray values were analyzed using Image J.

## 3. Statistical analysis

Using IBM SPSS Statistics 29 software, assuming normality or homogeneity of variance for all data, independent sample t-test, paired sample t-test, one-way ANOVA, and two-way ANOVA are performed. When the data does not follow a normal distribution, Mann Whitney U test is performed. Data were presented as mean ± standard error of mean, and Graphs were produced with the help of Adobe Illustrator 2022 and GraphPad Prism 9.

## 4. Results

### 4.1. Heroin withdrawal can cause significant withdrawal symptoms in mice

Previous studies have demonstrated that opioid antagonists like naloxone can precipitate significant withdrawal reactions [34,40]. However, further research is required to explore the natural withdrawal phenomena in heroin-dependent mice. To address this gap, we developed a 10-day heroin dependence model and subsequently assessed withdrawal symptoms, including behaviors such as standing positions, licking hair, face washing and bowel movements in mice 24 hours postabstinence [41,42].

In the context of withdrawal symptoms (Fig 1B-E), the heroin natural withdrawal group (H) exhibited a statistically significant reduction in the number of standing positions compared to the saline control group (S) (t (14) =3.331, $P = 0.005$, Fig 1B). Conversely, there were significant increases in the number of licking hair (Z (14) =−3.368, $P < 0.001$, Fig 1C), face-washing behaviors (t(14) =−6.860, $P < 0.001$, Fig 1D), and bowel movements (t (14) =−6.364, $P < 0.001$, Fig 1E). These findings indicate pronounced withdrawal symptoms in the heroin natural withdrawal mice.

### 4.2. Heroin withdrawal in mice is characterized by an increase in anxiety-like behaviors which may be associated with a reduction in glutamatergic neurons in the BLA, elevated c-Fos expression

Heroin withdrawal is associated with pronounced withdrawal symptoms in mice [34], suggesting the potential presence of anxiety and depression. To investigate alterations in anxiety and depression-related behaviors in C57BL/6J mice following a 7-day period of spontaneous withdrawal from heroin dependence, we induced heroin dependence in mice approximately 8 weeks old [35]. We subsequently assessed anxiety-like behaviors using the open field and elevated plus maze tests [36,37], and evaluated depression-like behaviors by measuring immobility time in the tail suspension and forced swim tests [39,43,44] (Fig 2A).

In the open field experiment (Fig 2B-E), compared with the S group mice, heroin withdrawal mice significantly reduced the time required to enter the time of the open field(t (14) =2.479, $P = 0.027$, Fig 2C), the distance in the center(t (14) = 3.202, $P = 0.006$, Fig 2D), and the center entries(t (14) = 2.640, $P = 0.019$, Fig 2E).

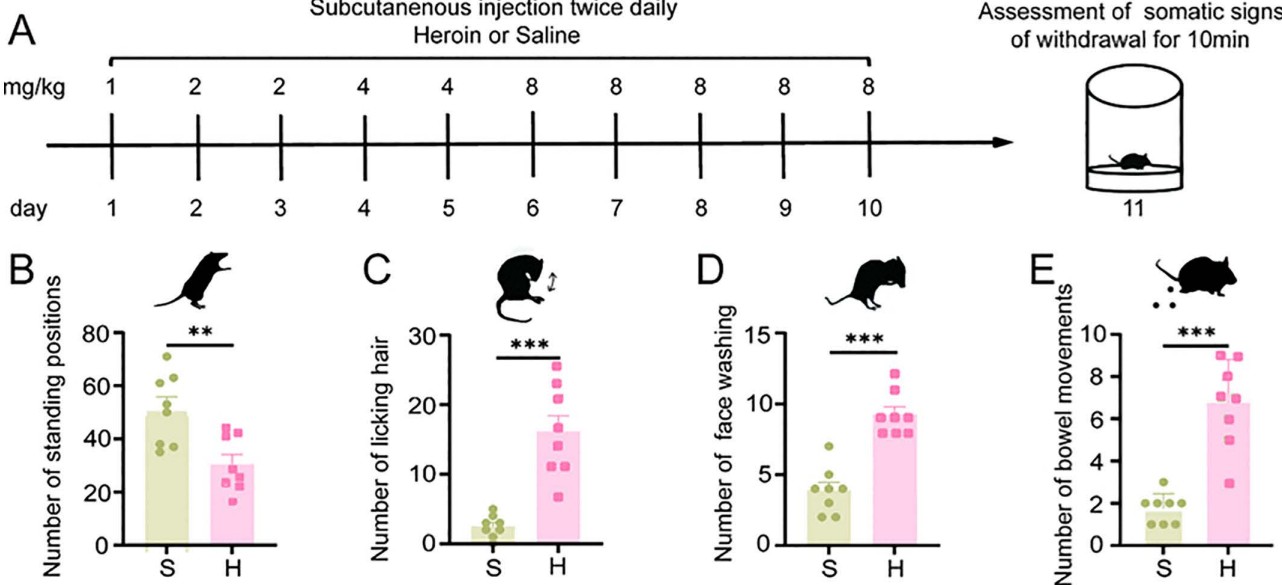

**Fig 1. Heroin withdrawal induces notable withdrawal reactions in mice. A**: Experimental flowchart for the establishment of a heroin withdrawal mouse model and assessment of withdrawal symptoms. **B-E**: Withdrawal symptoms (n = 8). Number of standing positions **(B)**, number of licking hair **(C)**, number of face washing **(D)** and number of bowel movements **(E)**. The results are presented as mean± standard error of the mean. No statistical significance is indicated by ns $P > 0.05$, and significant difference is indicated by * $P < 0.05$, ** $P < 0.01$, *** $P < 0.001$ indicates (Independent-sample t test).

In the elevated plus maze experiment (Fig 2F-I), heroin withdrawal mice exhibited reduced time spent in the open arms (t (14) = 4.604, $P < 0.001$, Fig 2G), fewer entries into the open arms (t(14) = 4.528, $P < 0.001$, Fig 2I), and decreased distance traveled in the open arms (Z(14) = −3.046, $P = 0.002$, Fig 2H).

The results from both the open field and elevated plus maze experiments indicate an increase in anxiety-like behaviors during heroin withdrawal.

In the forced swim test (Fig 2J), the immobility time of H group was significantly reduced compared to the S group (Z (14) = −3.363, $P < 0.001$).

In the tail suspension experiment (Fig 2K), the immobility time of H group mice was significantly lower than that of the S group (t (14) = 4.693, $P < 0.001$, Fig 2K).

The findings from the forced swim and tail suspension tests suggest a reduction in depression-like behaviors in mice during heroin withdrawal.

The results presented above suggest that during heroin withdrawal, mice predominantly exhibit symptoms of anxiety. This phenomenon may be associated with the risk of relapse during the withdrawal process from heroin.

Previous research has demonstrated that heightened anxiety levels may lead to increased c-Fos expression in brain regions such as the hippocampus, interpeduncular nucleus (IPN), and PFC [45–48]. In line with these findings, we conducted immunofluorescent staining (Fig 2L) to assess c-Fos protein expression in neurons within the BLA region of both S and H groups of mice. Our analysis revealed that the H group exhibited significantly elevated levels of c-Fos expression in the BLA compared to the S group (Z (10) = −2.756, $P = 0.006$, Fig 2M). Previous studies have identified the BLA region as being predominantly composed of glutamatergic neurons [49]. Consequently, we performed c-Fos and VGLUT1 immunofluorescent co-staining in the BLA region of both the H and S groups, which indicated that c-Fos expression was primarily localized to glutamatergic neurons (Z (18) = −3.780, $P < 0.001$, Fig 2N). Furthermore, we observed a reduction in the number of glutamatergic neurons in the H group (t (10) = −3.650, $P = 0.004$, Fig 2O). These findings suggest that in the BLA

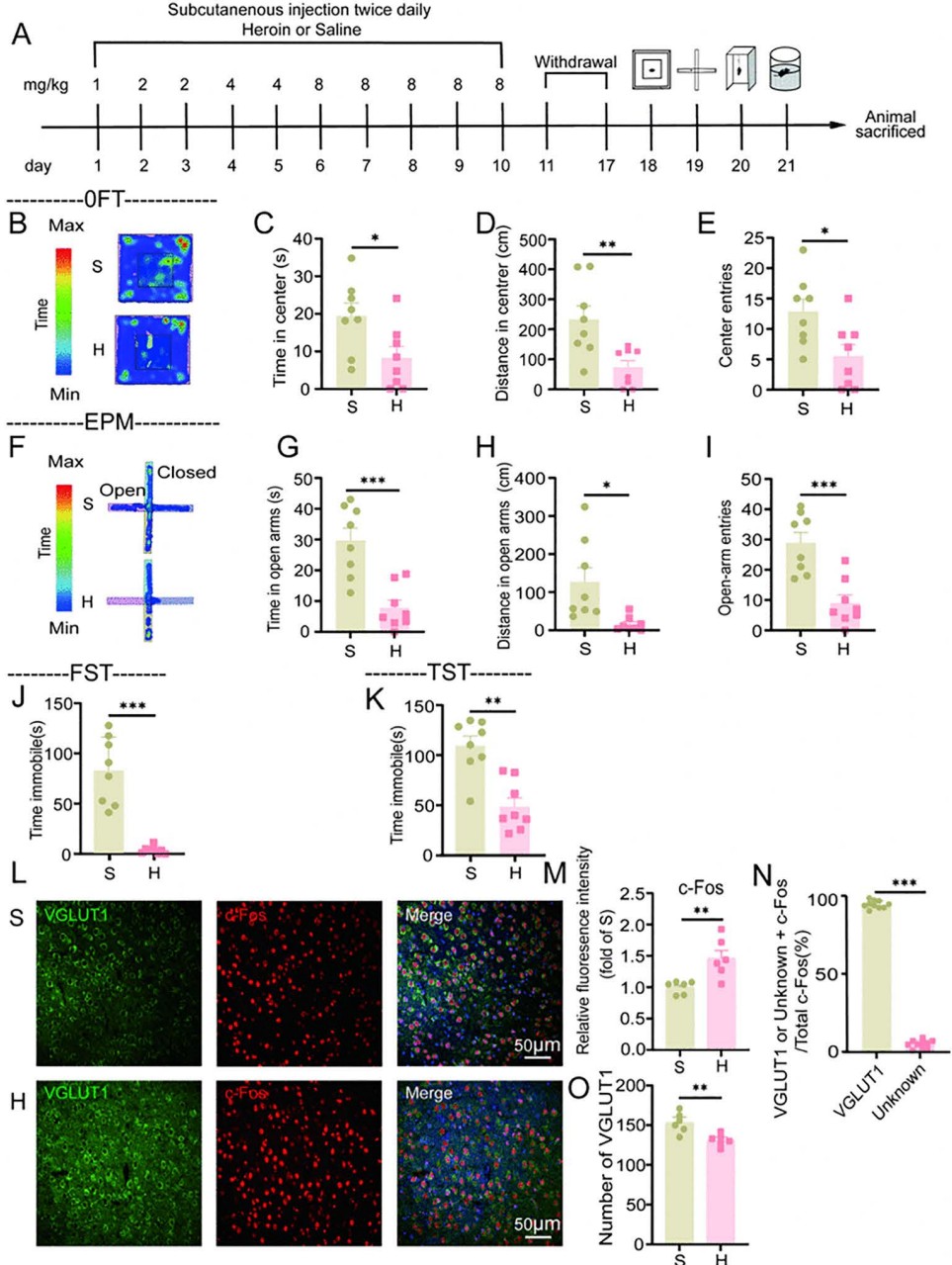

**Fig 2. Increased anxiety-related behaviors observed in mice undergoing heroin withdrawal. A**: Flowchart illustrating the modeling of heroin withdrawal in mice. **B-E**: Open field test (n = 8). Representative heatmaps of trajectories in the open field test **(B)**, time spent exploring the center area **(C)**, distance traveled in the center area **(D)**, and number of entries into the center area **(E)**. **F-I**: Elevated plus maze test (n = 8). Representative heatmaps of trajectories in the elevated plus maze test **(F)**, time spent in the open arms **(G)**, distance traveled in the open arms **(H)**, and number of entering the open arms **(I)**. **J**: Immobility time in the forced swim test (n = 8), **K**: Immobility time in the tail suspension test (n = 8). **L-O**: Immunofluorescence staining in the BLA brain region. Immunostaining for c-Fos/VGLUT1/DAPI **(L)**, average optical density of c-Fos immunofluorescence **(M)** (n = 6), proportion of glutamatergic neurons labeled with VGLUT1 and unidentified neurons expressing c-Fos among the total c-Fos-positive neurons labeled with VGLUT1 **(N)** (n = 10), and number of glutamatergic neurons **(O)** (n = 6). The results are presented as mean ± standard error of the mean. No statistical significance is indicated by ns $p > 0.05$, and significant difference is indicated by *$P < 0.05$, **$P < 0.01$, ***$P < 0.001$ indicates (Independent-sample t test and Mann-Whitney U test).

region of mice undergoing heroin withdrawal, there is a decrease in the number of glutamatergic neurons accompanied by an increase in c-Fos expression.

In summary, mice undergoing heroin withdrawal exhibited an increase in anxiety-like behaviors, which was accompanied by a reduction in the number of glutamatergic neurons within the BLA region of the brain and an elevation in c-Fos expression.

## 4.3. During the state of heroin dependence, no significant effects on anxiety and depressive behaviors were observed in mice. However, a reduction in glutamatergic neurons was identified in the BLA

Psychoactive substances, including cocaine, morphine, and other addictive drugs, have the capacity to modulate epigenetic mechanisms within specific brain regions, such as the PFC, NAc, VTA, and BLA [13,14,35,50–52]. These epigenetic modifications may intensify negative emotional states, such as anxiety and depression, in individuals with drug dependence, particularly during periods of dependence and withdrawal [3,4]. To investigate the alterations in anxiety and depression associated with heroin dependence, we established a heroin-dependent mouse model (Fig 3A). Behavioral evaluations were conducted 1–6 hours post-drug administration. Anxiety-like behaviors were assessed using the open field test and the elevated plus maze test, while depression-like behaviors were evaluated using the tail suspension test and the forced swim test.

In the open field test (Fig 3B-E), no significant differences were observed in the time spent in the center (t(14)=−1.166, $P=0.263$, Fig 3C), distance in center (t(14)=0.410, $P=0.688$, Fig 3D), and center entries (t(14)=1.463, $P=0.165$, Fig 3E) between the saline group (S') and the heroin-dependent group (H').

In the elevated plus maze experiment (Fig 3F-I), there were no significant differences in the time spent in the open arms (t(14)=−0.778, $P=0.449$, Fig 3G), distance in the open arms (t(14)=0.074, $P=0.942$, Fig 3H), and open-arms entries (t(14)=−0.581, $P=0.571$, Fig 3I) between the S' group and the H' group.

The findings from both the open field and elevated plus maze experiments demonstrated no significant alterations in anxiety-like behaviors among mice undergoing heroin withdrawal.

In the forced swim test (Fig 3J), no difference in immobility time was found between the H' group and the S' group (t(14)=1.338, $P=0.202$).

In the tail suspension test (Fig 3K), the immobility time was similar for both the H' group and the S' group (t(14)=0.397, $P=0.697$).

These findings from the forced swim and tail suspension tests suggest that there were no changes in depressive behaviors in mice during heroin withdrawal.

We conducted immunofluorescence staining in the BLA region of H group to explore whether there were negative changes in the brain area (Fig 3L). In the H group, the expression level of c-Fos in the BLA region did not exhibit a significant change when compared to the S group (t(10)= 0.836, $P=0.423$, Fig 3M). An analysis of c-Fos expression in glutamatergic and unknown neurons revealed that c-Fos was predominantly expressed in glutamatergic neurons (t(10) = −3.780, $P<0.001$, Fig 3N). Additionally, the number of excitatory neurons in the H' group was reduced (t(10) =2.724, $P=0.021$, Fig 3O).

In conclusion, the anxiety and depression-like behaviors of heroin dependent mice did not change, but the number of glutamatergic neurons in their BLA brain regions decreased. Heroin dependence reduces glutamatergic neurons in BLA without affecting anxiety and depression-like behaviors, due to the inhibitory effect of heroin reward.

## 4.4. There exists a neural circuit connecting the nucleus of the NTS and the BLA in mice. Withdrawal from heroin results in an increased signaling of GLP-1, which acts on glutamatergic neurons within the BLA through this pathway

Glucagon-like peptide-1 (GLP-1), recognized as a neurotrophic protein, is instrumental in various facets of neuronal growth and development within the central nervous system, with its effects being dependent on expression levels

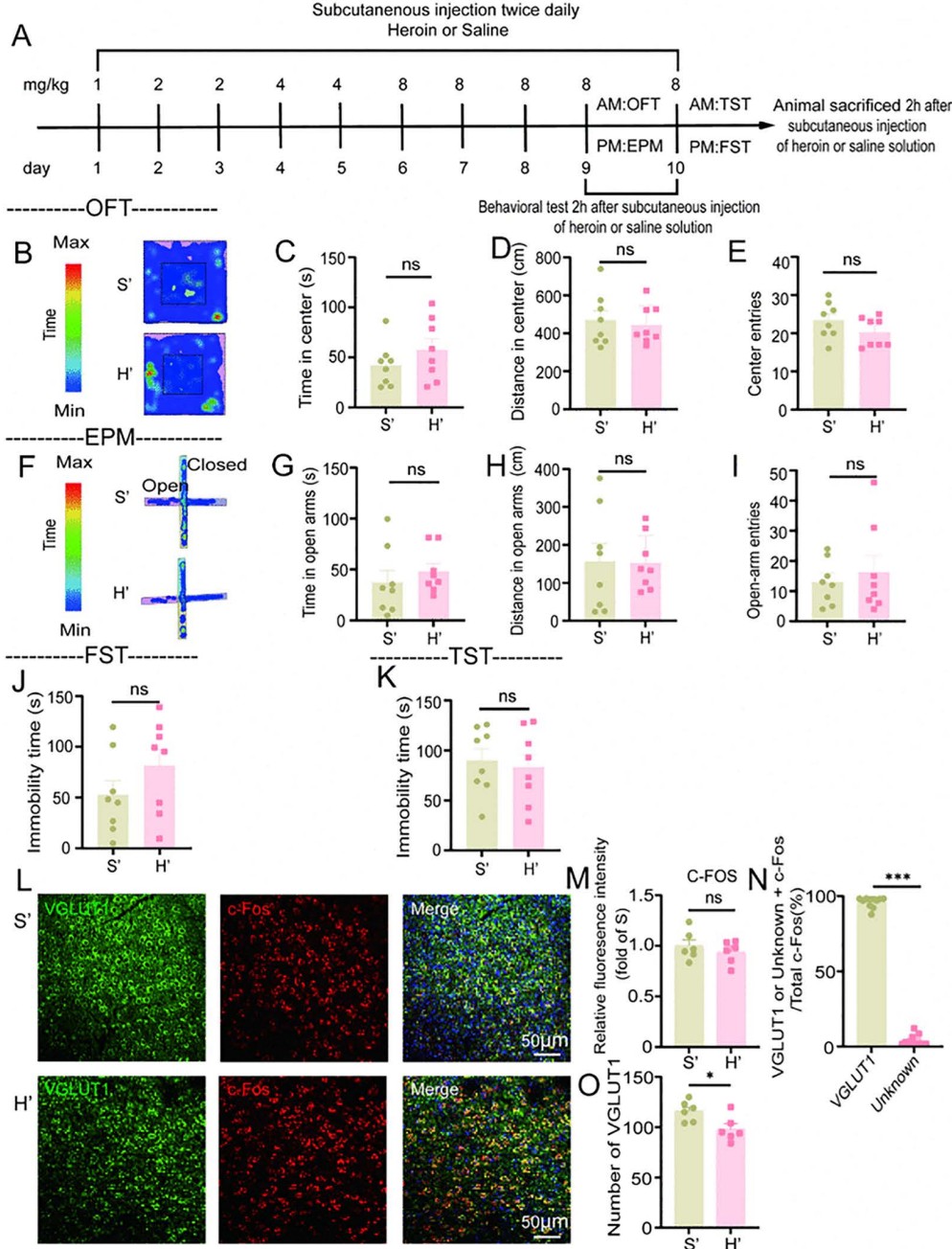

**Fig 3. Heroin dependence leads to a decrease in glutamatergic neurons within the BLA of mice. A**: Flow chart illustrating the heroin-dependent mice model; **B-E**: Open field test (n=8). Representative track heat map from the open field test **(B)**, time spent exploring the central area **(C)**, distance traveled in the central area **(D)**, and number of entries into the central area **(E)**; **F-I**: Elevated plus maze test (n=8). Representative track heat map from the elevated plus maze test **(F)**, time spent exploring the open arms **(G)**, distance traveled in the open arms **(H)**, and number of entries into the open arms **(I)**; **J**: Immobility time recorded in the forced swim test (n=8); **K**: Immobility time recorded in the tail suspension test (n=8); **L-O**: Immunofluorescence staining of the BLA. Immunofluorescence staining for c-Fos/VGLUT1/DAPI **(L)**, average optical density of c-Fos immunofluorescence **(M)** (n=6), proportion of glutamatergic and unidentified neurons expressing c-Fos among all c-Fos-positive neurons labeled with VGLUT1 **(N)** (n=10), and the number of glutamatergic neurons **(O)** (n=6); The results are presented as mean± standard error of the mean. No statistical significance is indicated by ns $P>0.05$, and significant difference is indicated by *$P<0.05$, ** $P<0.01$, ***$P<0.001$ indicates (Independent-samples t test and Mann-Whitney U test).

[20,22,53]. Its functions encompass anti-inflammatory and antioxidant activities, promotion of neuronal cell proliferation and differentiation, regulation of synaptic plasticity, and modulation of anxiety and depressive states [22,23,46,53]. To elucidate the neural pathways associated with GLP-1, we employed the retrograde tracing virus pAAV-hSyn-EGFP-2A-Cre-WPRE in the BLA region of the brain [6,20,54] (Fig 4A-B). This approach led to the identification of virus-infected neurons in the NTS region (Fig 4C), thereby demonstrating the presence of a neural pathway from the NTS to the BLA, corroborating findings from previous studies [55].

Although we have demonstrated the existence of a circuit between the NTS brain region and the BLA brain region, further research is needed to determine whether GLP-1 levels change during heroin withdrawal. For this, we collected samples from the NTS and BLA regions of mice in both H and S groups for WB analysis (Fig 4D-G). The findings indicated that, in comparison to the S group, there was a significant increase in the expression of GLP-1 in PPG neurons in the NTS of the H group (t (4) =−8.302, $P = 0.001$, Fig 4E). Additionally, the amount of GLP-1 projected to the BLA region (t (8) =−2.658, $P = 0.029$, Fig 4G) significantly increased. These results indicate that heroin withdrawal leads to upregulation of the GLP-1 signaling pathway in the NTS-BLA brain region of mice. Prior studies have shown that the upregulation of the GLP-1-related signaling pathway can cause increased anxiety-like behaviors [26,46], thus suggesting that the observed increased anxiety-like behaviors in heroin withdrawal mice may be related to the increased secretion of GLP-1 by PPG neurons in the NTS projecting to the BLA region, and the upregulated expression of GLP-1R within the BLA region.

To ascertain the primary type of neurons expressing GLP-1R, we conducted a double staining of glutamatergic neurons/GABAergic neurons and GLP-1R in the BLA region of the mice brain (Fig 4H-4I). Subsequent counting and statistical analysis revealed that the colocalization rate of glutamatergic neurons and GLP-1R was significantly higher than that of GABAergic neurons and GLP-1R (Z (6) =−2.882, $P = 0.004$, Fig 4I), demonstrating that GLP-1R is primarily distributed on glutamatergic neurons.

In summary, there exists a neural circuit connecting the NTS and the BLA regions of the brain in mice. Heroin withdrawal induces an increase in GLP-1 signaling within the BLA region as observed from the NTS. Notably, GLP-1R are predominantly expressed by glutamatergic neurons.

### 4.5. Intraperitoneal injection of exendin-4 reduces anxiety-like behaviors in heroin withdrawal mice

Exendin-4 is frequently employed in clinical settings as a pharmacological agent for weight reduction [20,56] and the management of diabetes [54]. To mitigate the risk of hypoglycemia in mice following Exendin-4 administration, we established two dosage gradients of Exendin-4, informed by a comprehensive literature review [26,32,57], specifically 2 μg/kg and 4 μg/kg. Blood samples were collected from the tail for glucose measurement one hour prior to and one hour following administration. It was observed that mice receiving 4 μg/kg Exendin-4 exhibited a significant reduction in blood glucose levels (t (5) = 5.396, $P = 0.003$, Fig 5A). To ascertain the impact of 2 μg/kg Exendin-4 on mouse body weight, the compound was administered bi-daily for a duration of seven consecutive days. In comparison to the saline control group, the 2 μg/kg Exendin-4 group did not demonstrate a statistically significant effect on body weight (t (6) = 1.566, $P = 0.148$, Fig 5B). Consequently, the selected dosage of Exendin-4 for this study is 2 μg/kg.

To investigate whether Exendin-4 can reduces anxiety in heroin withdrawal mice, we established a heroin withdrawal mouse model, followed by open field, elevated plus maze, forced swim, and tail suspension behavioral tests. Additionally, one week before the behavioral tests, half of the mice in the S group and the H group were intraperitoneally injected with Exendin-4 for 7 consecutive days [24,32] (Fig 5C).

To examine the potential anxiolytic effects of Exendin-4 in a murine model of heroin withdrawal, we developed a heroin withdrawal mouse model and subsequently conducted a series of behavioral assessments, including the open field test, elevated plus maze, forced swim test, and tail suspension test. Prior to these behavioral evaluations, a subset of mice from both the saline (S) and heroin (H) groups received intraperitoneal injections of Exendin-4 and saline for seven consecutive days (Fig 5C).

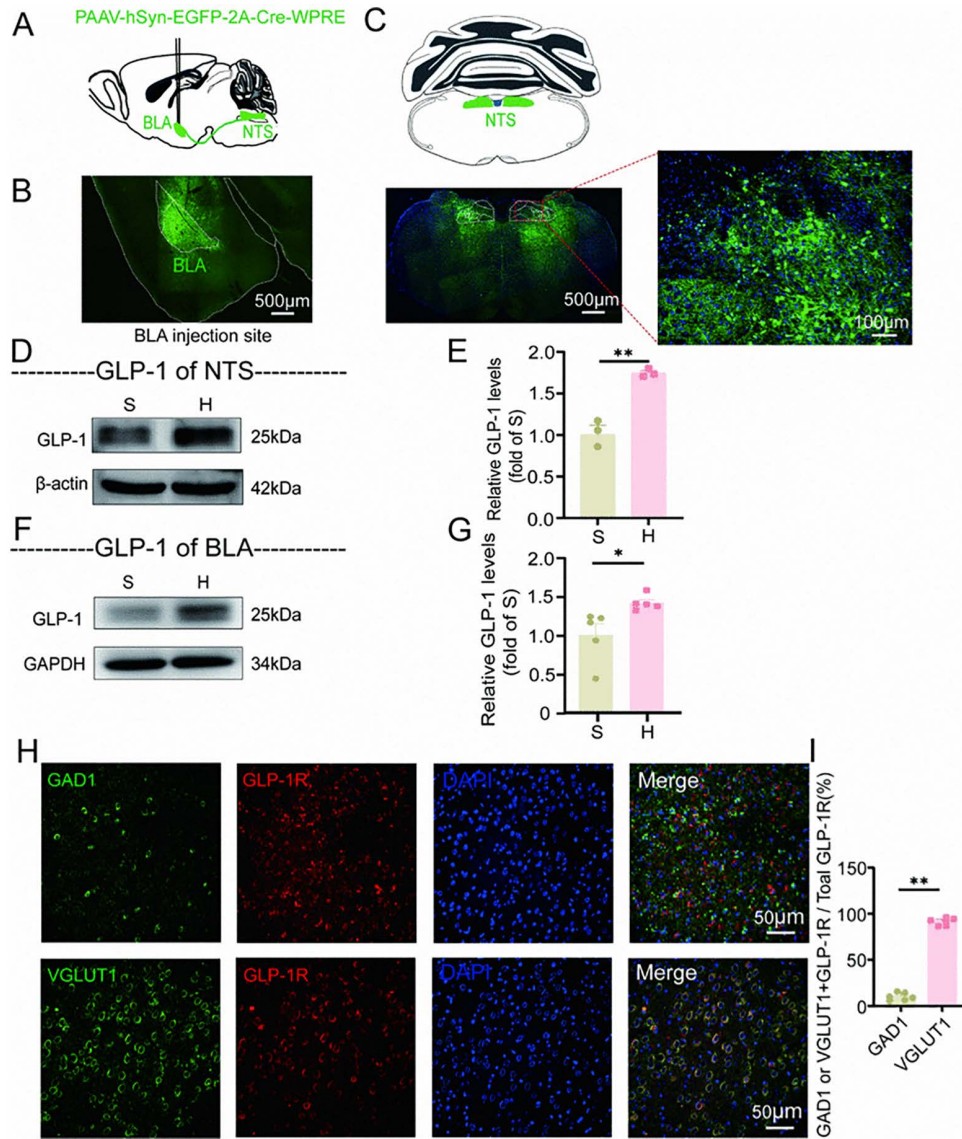

**Fig 4. Projection of NTS PPG neurons to the glutamatergic neurons of BLA in mice. A**: Schematic diagram of viral injection using pAAV-hSyn-EGFP-2A-Cre-WPRE;**B**: Stereotaxic injection site in the BLA; **C**: Morphological verification showing targeted NTS-BLA neurons in wild-type mice injected with pAAV-hSyn-EGFP-2A-Cre-WPRE; **D-E**: WB analysis of GLP-1 protein in the NTS. Representative images of total GLP-1 expression in the NTS **(D)** and grayscale values of GLP-1 protein **(E)** (n = 3), indicating increased GLP-1 expression in the NTS of mice undergoing heroin withdrawal; **F-G**: WB analysis of GLP-1 in the BLA. Representative images of total GLP-1 expression in the BLA **(F)**, grayscale values of GLP-1 protein (n = 5) **(G)**, demonstrating increased GLP-1 content in the BLA of mice during heroin withdrawal; **H-I**: Immunofluorescence staining. The number of glutamatergic/GABAergic neurons expressing GLP-1R **(J)** (n = 6). The results are presented as mean± standard error of the mean. No statistical significance is indicated by ns $P > 0.05$, significance levels indicated as $*P < 0.05$, $**P < 0.01$, and $***P < 0.001$ (Independent-samples t test).

In the open field test (Fig 5D-G), heroin withdrawal led to a decrease in the time spent by mice in the center zone ($F_{1,28} = 28.148$, $P < 0.001$, Fig 5E), while Exendin-4 intervention improved the time spent by Her + Ex-4 group mice in the center zone of the open field (Withdrawal×Exendin-4: $F_{1,31} = 25.121$, $P < 0.001$, Fig 5E) and the distance traveled (Withdrawal×Exendin-4: $F_{1,31} = 4.325$, $P = 0.047$, Fig 5F). The Exendin-4 treatment also led to a reduction in the time mice from the S group spent in the center zone ($F_{1,28} = 20.252$, $P < 0.001$, Fig 5E).

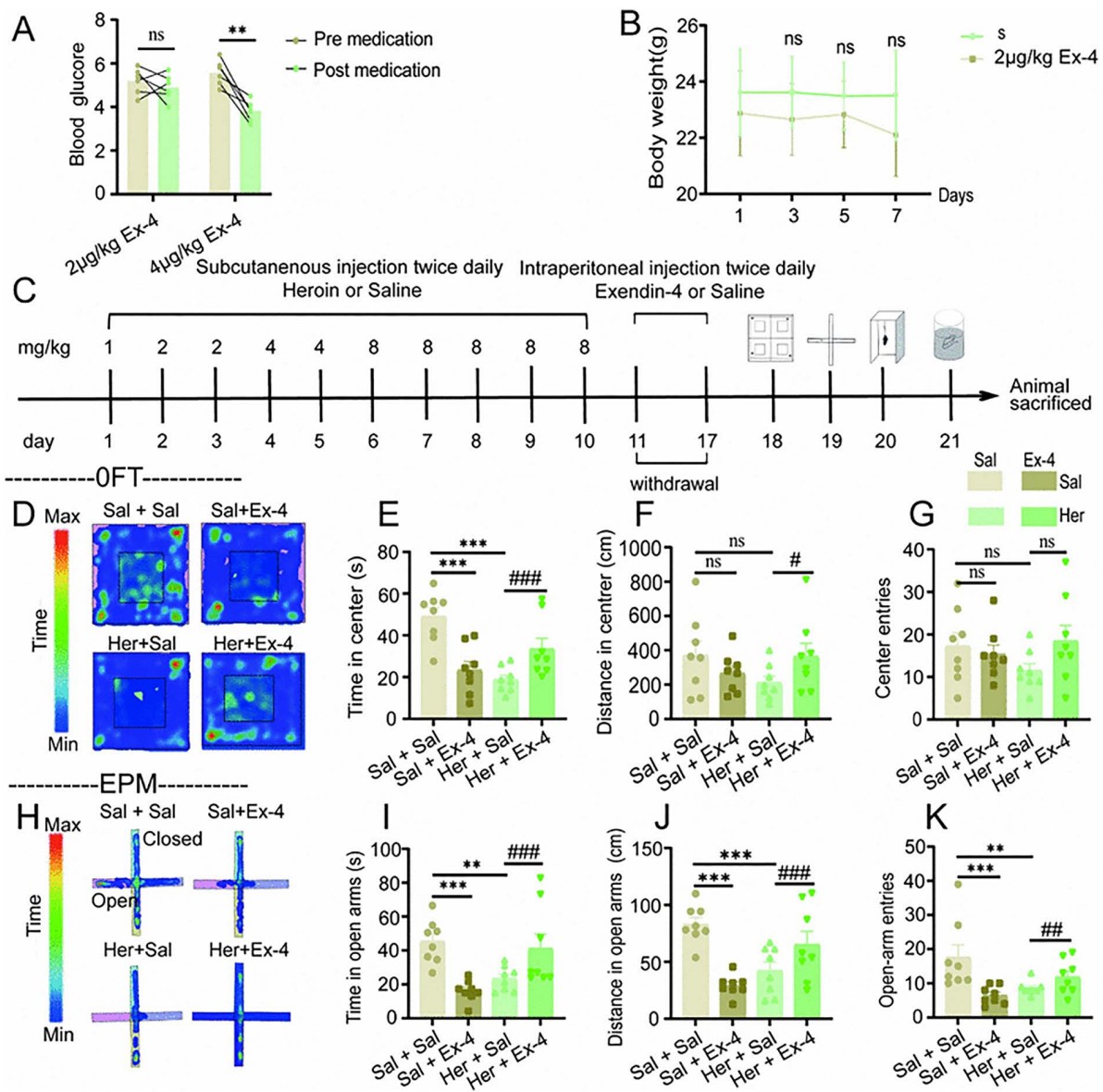

**Fig 5. Intraperitoneal administration of Exendin-4 significantly alleviated anxiety-like behaviors in mice undergoing withdrawal from heroin. A**: Represents blood glucose levels (n = 6); **B**: Reflects the body weight of mice treated with 2ug/kg Exendin-4 (n = 6); **C**: Illustrates the process of modeling heroin-dependent withdrawal in mice; **D-G**: Open field test (n = 8). Representative track heat map in the open field test **(D)**, time spent exploring in the center area **(E)**, distance traveled in the center area **(F)**, number of entries into the center area **(G)**; **H-K**: Elevated plus maze test (n = 8). Representative track heat map in the elevated plus maze test **(H)**, time spent exploring in the open arms **(I)**, distance traveled in the open arms **(J)**, number of entries into the open arms **(K)**; The results are presented as mean ± standard error of the mean. No statistical significance is indicated by ns $P > 0.05$, significance levels indicated as #/*$P < 0.05$, ##/**$P < 0.01$, and ###/***$P < 0.001$ (Paired sample t-test, independent sample t-test and two factor analysis of variance).

In the elevated plus maze (Fig 5H-K), heroin withdrawal reduced the time spent ($F_{1,28} = 9.207$, $P = 0.005$, Fig 5I), the distance traveled ($F_{1,28} = 14.078$, $P < 0.001$, Fig 5J), and the number of entries ($F_{1,28} = 9.214$, $P = 0.005$, Fig 5K) in the open arms by mice, while Exendin-4 intervention improved the time spent (Withdrawal×Exendin-4: $F_{1,31} = 21.124$, $P < 0.001$, Fig 5I), the distance traveled (Withdrawal×Exendin-4: $F_{1,31} = 25.154$, $P < 0.001$, Fig 5J), and the number of entries

(Withdrawal×Exendin-4: $F_{1,31} = 11.753$, $P = 0.002$, Fig 5K) in the open arms by Her + Ex-4 group of mice. However, Exendin-4 also decreased the time spent ($F_{1,28} = 16.304$, $P < 0.001$, Fig 5I), the distance traveled ($F_{1,28} = 24.735$, $P < 0.001$, Fig 5J), and the number of entries ($F_{1,28} = 14.078$, $P < 0.001$, Fig 5K) in the open arms by mice in the Sal + Ex-4 group.

The outcomes of the open field and elevated plus maze experiments indicated that the anxiety-like behaviors exhibited by mice in both the Sal + Ex-4 group and the Her + Sal group were comparable. This similarity suggests that the alteration in anxiety-like behaviors observed in the Her + Sal group is linked to the activation of the GLP-1-related signaling pathway, corroborating our previous findings. Moreover, administration of Exendin-4 resulted in a significant reduction in anxiety-like behaviors in mice within the Her + Ex-4 group.

We performed Western blot analysis (Fig 6A-F) and immunofluorescence staining (Fig 6G-I) on the BLA region of the Sal + Sal, Sal + Ex-4, Her + Sal, and Her + Ex-4 mouse groups. According to the literature, TrkB and BDNF are widely recognized as markers indicative of anxiety-like behaviors in mice [46,58]. These markers are also associated with CREB, thereby playing a role in the regulation of neuronal proliferation and differentiation, as well as influencing the expression of c-Fos [59].

The protein immunoblotting results revealed that heroin withdrawal significantly elevated the expression levels of BDNF ($F_{1,20} = 14.261$, $P = 0.001$, Fig 6C), TrkB ($F_{1,16} = 9.264$, $P = 0.008$, Fig 6D), GLP-1R ($F_{1,12} = 17.539$, $P = 0.001$, Fig 6B), GLP-1 ($F_{1,12} = 10.874$, $P = 0.006$, Fig 6F), and CREB ($F_{1,24} = 5.920$, $P = 0.023$, Fig 6E) in the Her + Sal group compared to the Sal + Sal group. However, intraperitoneal injection of Exendin-4 enhanced the expression levels of TrkB (Withdrawal× Exendin-4: $F_{1,19} = 20.976$, $P < 0.001$, Fig 6D), BDNF (Withdrawal×Exendin-4: $F_{1,23} = 42.259$, $P < 0.001$, Fig 6C), and GLP-1 (Withdrawal ×Exendin-4: $F_{1,15} = 19.198$, $P < 0.001$, Fig 6F) in the Her + Ex-4 group. In contrast, Exendin-4 also increased the expression levels of BDNF ($F_{1,20} = 27.642$, $P < 0.001$, Fig 6C), TrkB ($F_{1,16} = 15.307$, $P = 0.001$, Fig 6D), GLP-1R ($F_{1,12} = 11.106$, $P = 0.006$, Fig 6B), GLP-1 ($F_{1,12} = 9.553$, $P = 0.009$, Fig 6F), and CREB ($F_{1,24} = 3.342$, $P = 0.08$, Fig 6E) in the Sal + Ex-4 mice. These findings indicate that the increased anxiety-like behaviors observed in heroin withdrawal mice is correlated with elevated protein expression levels of GLP-1, TrkB, BDNF, and other associated proteins. In contrast, the molecular protein alterations in Sal + Ex-4 mice remain consistent. The reduction in anxiety-like behaviors seen in Her + Ex-4 mice may be attributed to a decrease in the protein expression of TrkB, BDNF, and GLP-1.

Finally, we conducted immunofluorescent staining of the BLA brain region in mice. Significant variations in the count of glutamatergic neurons were noted between the Sal + Sal, Sal + Ex-4, Her + Sal, and Her + Ex-4 groups. ($F_{(3,23)} = 37.327$, $P < 0.001$, Fig 6I). Post hoc comparisons revealed that the number of glutamatergic neurons increased in the Sal + Ex-4 group ($P < 0.001$), decreased in the Her + Sal group ($P < 0.001$), and increased in the Her + Ex-4 group compared to the Her + Sal group ($P < 0.001$). Differences in c-Fos expression were also noted among the Sal + Sal, Sal + Ex-4, Her + Sal, and Her + Ex-4 groups ($F_{(3,23)} = 66.457$, $P = 0.003$, Fig 6H). Post hoc comparisons showed that c-Fos expression increased in the Sal + Ex-4 ($P = 0.012$), Her + Sal ($P < 0.001$), and Her + Ex-4 ($P = 0.077$) groups compared to the Sal + Sal group. Additionally, c-Fos expression decreased in the Her + Ex-4 group compared to the Her + Sal group ($P = 0.024$). These results suggest that Exendin-4 can increase the number of glutamatergic neurons in the BLA brain region and is associated with a decrease in c-Fos expression following the cessation of Her + Ex-4 treatment.

In conclusion, the alterations in anxiety-like behaviors observed in the Her + Sal group mice are associated with neural repair mechanisms facilitated by the upregulation of the GLP-1 signaling pathway. These effects can be replicated through the intraperitoneal administration of Exendin-4 in the Sal + Ex-4 group. Conversely, Exendin-4 appears to mitigate anxiety-like behaviors in the Her + Ex-4 group, potentially due to an increase in the number of glutamatergic neurons within the BLA region of these mice. This increase may lead to a reduction in the projection of the neural repair protein GLP-1, subsequently downregulating the expression of c-Fos, TrkB, and BDNF.

## 5. Discussion

The basolateral amygdala (BLA) is integral to drug relapse research, particularly concerning the formation of drug-related emotional memories and alterations in neural plasticity. The BLA facilitates relapse behaviors by encoding the intense

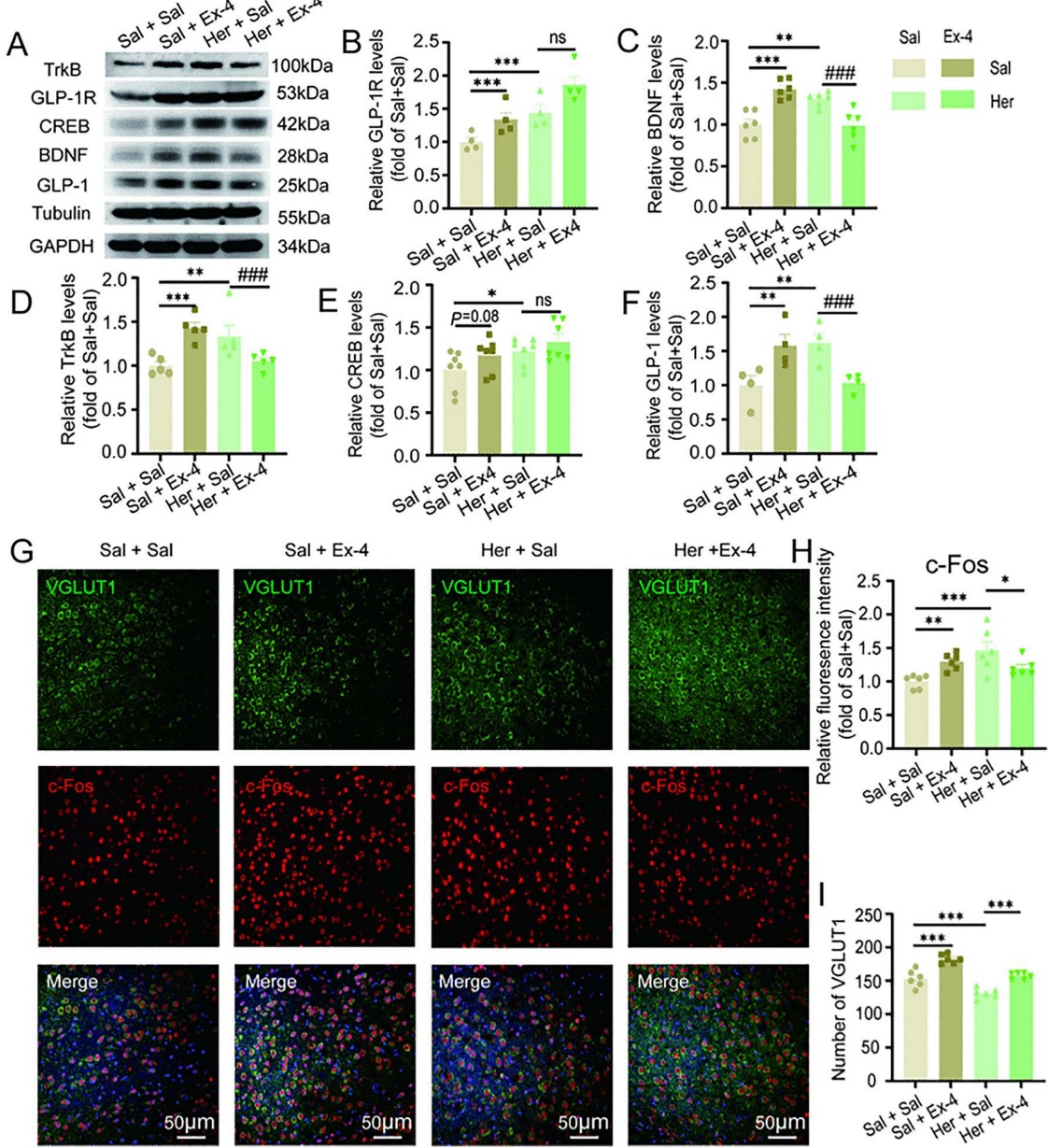

**Fig 6. Intraperitoneal administration of Exendin-4 has been demonstrated to enhance the population of glutamatergic neurons in the BLA of mice undergoing heroin withdrawal.** This effect is accompanied by a reduction in c-Fos protein expression. **A-F**: Western blot analysis of GLP-1R, TrkB, CREB, GLP-1, and BDNF protein in the BLA. Representative immunoblot images of total expression of GLP-1R, TrkB, CREB, GLP-1, and BDNF in the BLA **(A)**, grayscale value of GLP-1R protein **(B)** (n = 4), grayscale value of BDNF protein **(C)** (n = 6), grayscale value of TrkB protein **(D)** (n = 5), grayscale value of CREB protein **(E)** (n = 7), grayscale value of GLP-1 protein **(F)** (n = 4). Increased expression of GLP-1R, TrkB, CREB, GLP-1, and BDNF was observed in the BLA of the Her + Sal and Sal + Sal groups mice. The expression of GLP-1, TrkB, and BDNF in the BLA was decreased in the Her + Ex-4 group mice compared to the Her + Sal group. **G-I**: Immunofluorescence staining. Immunofluorescence staining of c-Fos/VGLUT1/ DAPI in the BLA **(G)**, average light density of c-Fos immunofluorescence **(H)**, number of glutamatergic neurons **(I)** (n = 6). The results are presented as mean ± standard error of the mean. No statistical significance is indicated by ns $P > 0.05$, significance levels indicated as #/*$P < 0.05$, ##/**$P < 0.01$, and ###/***$P < 0.001$ (One factor analysis of variance and two factor analysis of variance).

emotional experiences associated with drug use and the memories linked to drug-related cues. Modulation of BLA neurons can influence drug-seeking behaviors [60–64], positioning the BLA as a significant target for developing novel therapeutic agents for relapse. Recent studies have suggested that the VTA projection to the BLA is implicated in the regulation of anxiety behaviors [6]. Moreover, inflammation and the activity of the noradrenergic system within BLA regions can affect anxiety and depressive behaviors in mice [11,12]. In this study, we identified a neural circuit in male mice that compensates for the reduction in the number of glutamatergic neurons in the BLA due to heroin dependence, extending from the NTS to the BLA. This circuit is associated with the activation of glutamatergic neurons in the BLA, resulting in anxiety-like behaviors in male mice undergoing heroin withdrawal. Notably, a seven-day regimen of intraperitoneal administration of the GLP-1 analog Exendin-4 has been observed to attenuate anxiety-like behaviors in mice undergoing heroin withdrawal. This effect may be associated with the restoration of glutamatergic neuron populations within the BLA region of the brain, subsequently resulting in a reduction in the projection of the neural repair protein GLP-1 (Fig 7).

We conducted retrograde tracing by administering the pAAV-hSyn-EGFP-2A-Cre WPRE virus into the BLA, which resulted in the observation of virus-infected neurons within the NTS. Furthermore, through immunofluorescence staining, we identified GLP-1 protein in both the NTS and BLA. Our results demonstrate that GLP-1 protein is synthesized in the NTS of mice, whereas the BLA expresses the GLP-1R and receives GLP-1 projections, thereby confirming the presence of a neural circuit between these brain regions and highlighting the role of the GLP-1 signaling pathway [20,55]. Additionally, immunofluorescence staining for GLP-1R, along with markers for glutamatergic and GABAergic neurons, revealed that GLP-1R is predominantly expressed in glutamatergic neurons within the BLA.

As a neural repair protein, GLP-1 exerts anti-inflammatory and antioxidant effects, contributing to the protection of neurons and the promotion of neuronal cell proliferation and differentiation within the central nervous system [21,22]. Conversely, heroin dependence adversely impacts multiple brain regions, including the PFC, Hip, BLA, NAc and VTA. This results in alterations in neurotransmitter secretion, neuronal plasticity, inflammatory responses, and structural changes [8,13,51,65]. The mechanisms underlying these effects are particularly intriguing. A study conducted by Sofia Bouhlal and Gustavo A. demonstrated a reduction in GLP-1 levels one hour following cocaine administration [21,66,67], indicating that the neural repair function of GLP-1 may be inhibited during drug dependence, a phenomenon similarly observed in Alzheimer's disease [22,53].

We conducted immunostaining on the BLA of mice with heroin dependence and identified a reduction in the population of glutamatergic neurons within this brain region. Despite this neuronal alteration, the anxiety and depression-like behaviors of heroin dependent mice remained unchanged, possibly due to the rewarding effect of heroin inhibiting their anxiety [7,66,67]. After a 24-hour natural heroin withdrawal period, mice showed significant withdrawal symptoms, with a decrease in standing number and an increase in standing, licking, and defecation number. After a 7-day withdrawal period, the mice demonstrated an increase in anxiety-like behaviors and a decrease in depression-like behaviors, suggesting that anxiety related to heroin relapse was predominant at this stage of withdrawal. To further investigate these findings, we performed Western blotting and immunofluorescence staining on the BLA of mice after 7 days of heroin withdrawal. These analyses showed that GLP-1 levels were elevated in NTS, GLP-1 R and c-Fos expression were increased, and glutamatergic neurons remained at low levels in BLA.

Previous studies have indicated that the GLP-1 signaling pathway is involved in regulating c-Fos expression and promoting neuronal proliferation [22,33,53], and its enhanced transmission leads to an increase in anxiety-like behaviors [26,46], while the expression of c-Fos in related brain regions is elevated [45,68,69]. Based on our findings, the 7-day heroin withdrawal period in mice resulted in upregulation of the GLP-1 signaling pathway in NTS-BLA, which led to an increase in c-Fos expression without increasing the number of glutamatergic neurons in BLA.

We propose the hypothesis that heroin dependence results in a reduction of glutamatergic neurons within the BLA. Following withdrawal, GLP-1 is implicated in neural repair processes within the central nervous system, leading to an upregulation of c-Fos expression in glutamatergic neurons located in the BLA. An increase in the population of glutamatergic neurons in the BLA may subsequently restore GLP-1 secretion to normal levels in these mice.

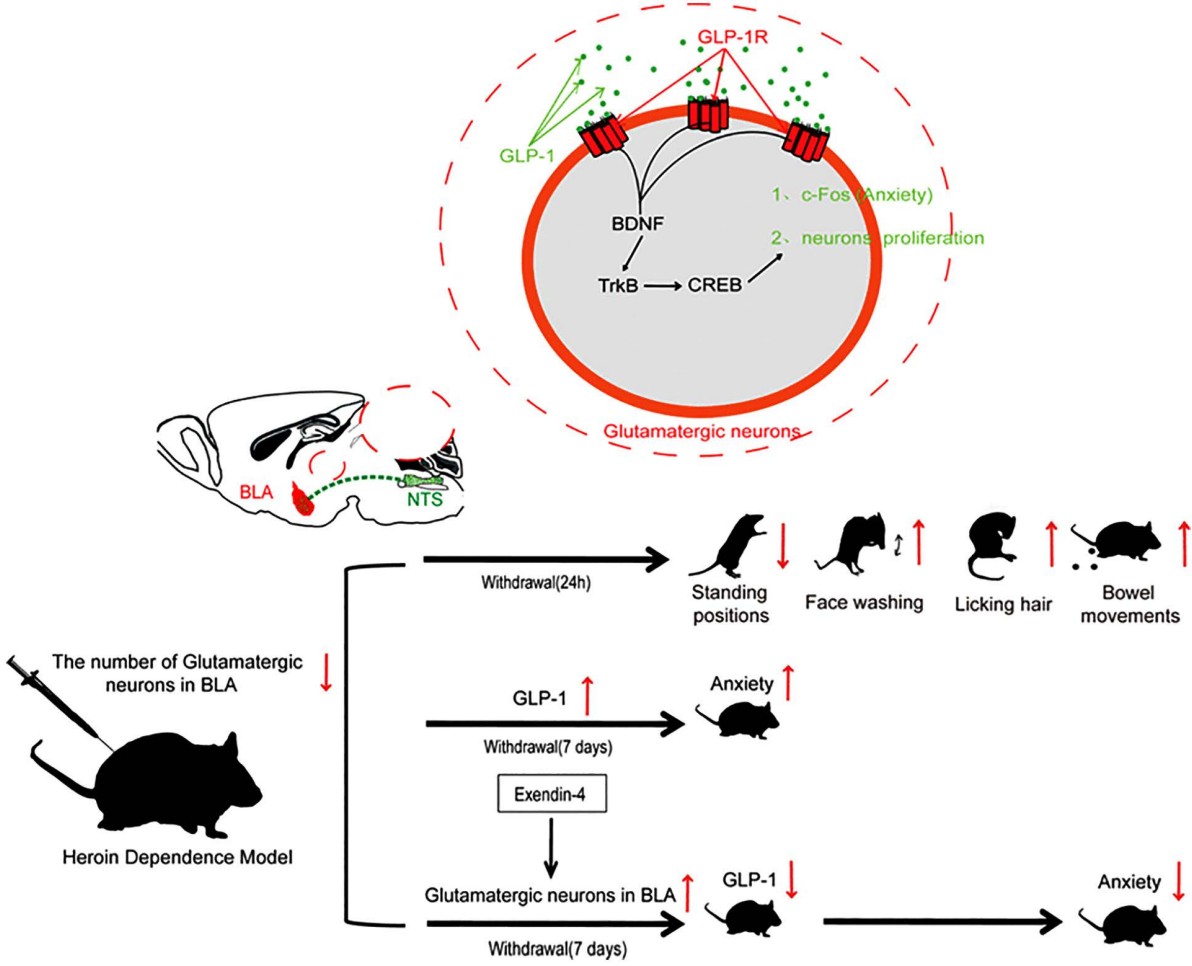

**Fig 7. Heroin dependence has been shown to negatively impact the BLA, resulting in a reduction in the population of glutamatergic neurons.** After withdrawal from heroin dependent mice, significant withdrawal symptoms appear. During this period, the synthesis and secretion of GLP-1 increase in NTS pre glucagon neurons, targeting BLA glutamatergic neurons, promoting the synthesis of BDNF, TrkB, CREB, and c-Fos proteins, facilitating neural repair, and leading to increased anxiety. However, upregulation of GLP-1 signaling does not restore the number of glutamatergic neurons in BLA. Our study found that the administration of Exendin-4 can enhance the neural repair effect of GLP-1. Once the glutamatergic neuron population in BLA recovers, GLP-1 projection to BLA subsequently decreases, which is associated with a reduction in anxiety-like behaviors.

The factors contributing to drug relapse are multifaceted and intricate, encompassing sleep disorders, anxiety, depression, fear, and social avoidance, among others. The prolonged duration of withdrawal symptoms and the enduring distress experienced by individuals undergoing drug withdrawal [3,4,70] contribute to the complexity of relapse causation [3,71–73]. Research indicates that natural withdrawal in opioid-dependent mice is associated with heightened anxiety and depression-like behaviors [3], which serve as significant motivators for drug relapse [3,7,74]. GLP-1 has demonstrated efficacy in addressing drug relapse by markedly reducing the self-administration of substances such as heroin, alcohol, and cocaine, as well as preventing relapse in murine models [29–33,57]. However, the underlying mechanisms by which GLP-1 alleviates relapse-associated anxiety remain unresolved. According to studies by Guoyong Ren and Han Weina, the sustained administration of GLP-1 analogs significantly diminishes depression and anxiety-like behaviors in mice [75,76], suggesting that the anxiolytic effects of GLP-1 therapy are linked to its long-term use.

We investigated the mechanism by which GLP-1 treatment affects anxiety in mice undergoing heroin withdrawal, utilizing intraperitoneal injections of the GLP-1 analogue Exendin-4 over a period of seven days [24,26]. Our findings indicate that Exendin-4 administration ameliorated anxiety symptoms in heroin-withdrawn mice [75,76]. This effect may be attributed to an increase in the number of glutamatergic neurons within the BLA, which subsequently led to a reduction in the projection of the neural repair protein GLP-1 in the NTS-BLA pathway [46], and a downregulation of c-Fos expression in the BLA. Conversely, in the saline-treated control group, a seven-day Exendin-4 intervention resulted in heightened anxiety and reduced depression [22,26]. This phenomenon may be associated with an increase in the number of gluta-matergic neurons, as well as elevated levels of c-Fos, GLP-1 expression, and GLP-1 receptor activity in the BLA. Further investigation is warranted to elucidate the underlying reasons for the elevated GLP-1 expression and receptor activity observed in the BLA of the saline control group following the cessation of Exendin-4 treatment. Our observations indicate that Exendin-4 treatment ameliorated anxiety symptoms in mice undergoing heroin withdrawal, yet paradoxically exacer-bated anxiety-like behaviors in the saline control group.

In summary, GLP-1, a neural repair protein, plays a critical role in modulating anxiety-like behaviors in mice undergoing heroin withdrawal. This protein is projected from the NTS to the BLA, where it enhances c-Fos expression in glutamater-gic neurons. Notably, heroin dependence is associated with a reduction in glutamatergic neurons within the BLA. During heroin withdrawal, this signaling pathway is activated as a compensatory mechanism to restore the glutamatergic neuron population in the BLA. However, the endogenous secretion of GLP-1 in mice is insufficient to significantly increase the number of glutamatergic neurons in this region, potentially exacerbating anxiety-like behaviors. This limitation can be addressed through the administration of Exendin-4 via intraperitoneal injection.

## 6. Limitations of the research

At present, the precise roles of optogenetics and chemogenetics within the NTS-BLA circuit remain unverified, warrant-ing further investigation into their specific functions. Our research has demonstrated that intraperitoneal administration of Exendin-4 can enhance GLP-1-mediated neural repair in the BLA brain regions. However, it is crucial to acknowledge that this administration may also affect the inflammatory response in BLA and non BLA brain regions or improve heroin withdrawal symptoms, potentially alleviating anxiety associated with heroin withdrawal [12,75]. The small sample size of WB for neural repair related proteins in these brain regions reduces its credibility. In addition, previous studies have shown that administration of Exendin-4 may increase patients' anxiety, which may exacerbate the anxiety of heroin withdrawal patients after medication, making it difficult for them to tolerate this anxiety and give up continuing medication or develop into depression, which may limit the development of GLP-1 analogs [27]. Addressing this side effect would substantially improve the feasibility of employing GLP-1 analogs for opioid withdrawal treatment. Additionally, the animal model uti-lized in our study represents a chronic heroin dependence model, and its relevance to heroin addiction requires further investigation.

## Supporting information

**S1 Raw Image. Raw images of WB.**
(PDF)

**S1 Data. Raw data of withdrawal symptoms.**
(XLSX)

**S2 Data. Raw data of heroin withdrawal mice.**
(XLSX)

**S3 Data. Raw data of heroin dependence mice.**
(XLSX)

**S4 Data. Raw data of NTS-BLA of heroin withdrawal mice.**
(XLSX)

**S5 Data. Raw data on the behavioral effects of Ex-4 on heroin withdrawal mice.**
(XLSX)

**S6 Data. Raw data on effect of Ex-4 on the protein expression in BLA of heroin withdrawal mice.**
(XLSX)

**S1 File. Experimental method.**
(DOCX)

## Author contributions

**Conceptualization:** Yang Xiang, Xiaowei Yan, Chunlu Li, Cancan Luo, Yingqi Xiong, Yan Hong.

**Data curation:** Baijuan Xia, Yang Xiang, Dan Yin, Yixin Li.

**Formal analysis:** Dan Yin, Yixin Li.

**Funding acquisition:** Baijuan Xia, Rongrong Li, Yixin Li.

**Investigation:** Yang Xiang, Rongrong Li, Cancan Luo.

**Methodology:** Yang Xiang, Xiaowei Yan, Rongrong Li, Chunlu Li, Dan Yin, Cancan Luo, Yan Hong.

**Project administration:** Baijuan Xia, Yang Xiang, Yan Hong.

**Resources:** Xiaowei Yan, Dan Yin, Yixin Li.

**Software:** Baijuan Xia, Yang Xiang, Xiaowei Yan, Rongrong Li, Chunlu Li, Cancan Luo, Yingqi Xiong.

**Validation:** Baijuan Xia.

**Writing – original draft:** Yang Xiang, Xiaowei Yan, Yingqi Xiong.

**Writing – review & editing:** Baijuan Xia, Yang Xiang, Rongrong Li, Chunlu Li, Yan Hong, Yixin Li.

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
