## [Decision Letter · Decision Letter 0]

12 Nov 2025

Dear Dr. Xia,

Thank you for submitting your manuscript to PLOS ONE. After careful consideration, we feel that it has merit but does not fully meet PLOS ONE’s publication criteria as it currently stands. Therefore, we invite you to submit a revised version of the manuscript that addresses the points raised during the review process.

This manuscript examines GLP-1 signaling in anxiety during heroin withdrawal but suffers from conceptual, methodological, and interpretative weaknesses that limit its impact. The introduction contradicts later claims about glutamatergic neuron loss and anxiety, and behavioral interpretations may conflate hyperactivity with reduced depression. Exclusive use of male mice introduces sex bias, and mechanistic claims that Exendin-4 “restores glutamatergic neurons” lack direct evidence of neurogenesis or synaptic repair. Statistical reporting is error-prone, with implausible t-values and uncorrected multiple comparisons, while small and inconsistent sample sizes undermine power. The viral tracing experiments lack validation and pharmacological controls are insufficient, with inconsistent drug routes, single-dose design, and no verification of withdrawal models. Overall, while the study addresses a relevant neurobiological question, its conclusions overreach the data and require substantial clarification and methodological strengthening before consideration for publication.

We look forward to receiving your revised manuscript.

Kind regards,

Mohammad Sarif Mohiuddin

Academic Editor

PLOS ONE

2. To comply with PLOS One submissions requirements, in your Methods section, please provide additional information regarding the experiments involving animals and ensure you have included details on (1) methods of sacrifice, (2) methods of anesthesia and/or analgesia, and (3) efforts to alleviate suffering.

[This initiative received financial support from the National Natural Science Foundation of China [Grant numbers: 81960253,82160268 &82201653], Guizhou Provincial Science and Technology Program Project (Grant numbers QianKeHeJiChu ZK (2024) General 123), National Natural Science Foundation Cultivation Project  [Grant number 21NSFCP37], and Key Laboratory for Research on Autoimmune Diseases of Higher Education Schools in Guizhou Province (Qianjiaoji [2023] 016).].

6. Please include a separate caption for each figure in your manuscript.

Additional Editor Comments:

This manuscript examines GLP-1 signaling in anxiety during heroin withdrawal but suffers from conceptual, methodological, and interpretative weaknesses that limit its impact. The introduction contradicts later claims about glutamatergic neuron loss and anxiety, and behavioral interpretations may conflate hyperactivity with reduced depression. Exclusive use of male mice introduces sex bias, and mechanistic claims that Exendin-4 “restores glutamatergic neurons” lack direct evidence of neurogenesis or synaptic repair. Statistical reporting is error-prone, with implausible t-values and uncorrected multiple comparisons, while small and inconsistent sample sizes undermine power. The viral tracing experiments lack validation and pharmacological controls are insufficient, with inconsistent drug routes, single-dose design, and no verification of withdrawal models. Overall, while the study addresses a relevant neurobiological question, its conclusions overreach the data and require substantial clarification and methodological strengthening before consideration for publication.

Reviewers' comments:

Reviewer's Responses to Questions

**Comments to the Author**

1. Is the manuscript technically sound, and do the data support the conclusions?

Reviewer #1: Yes

Reviewer #2: Yes

2. Has the statistical analysis been performed appropriately and rigorously?

Reviewer #1: Yes

Reviewer #2: Yes

3. Have the authors made all data underlying the findings in their manuscript fully available?

Reviewer #1: Yes

Reviewer #2: Yes

4. Is the manuscript presented in an intelligible fashion and written in standard English?

Reviewer #1: Yes

Reviewer #2: Yes

Reviewer #1: 1. Conceptual contradiction

The introduction claims heroin dependence reduces glutamatergic neurons “without affecting anxiety or depression,” but the results later attribute increased anxiety during withdrawal to this reduction. This is inconsistent and requires clarification.

2. Sex bias in study design

Only male mice were used. Given well-documented sex differences in addiction and anxiety behaviors, this omission limits translational value.

3. Misinterpretation of depression-like behaviors

Reduced immobility in forced swim and tail suspension tests is interpreted as “reduced depression-like behavior,” but this could also reflect withdrawal-induced hyperactivity or agitation. The interpretation is misleading.

4. Mechanistic overstatement

The discussion repeatedly states Exendin-4 “restores glutamatergic neurons” and provides “neural repair,” yet no direct evidence of neurogenesis or synaptic remodeling was provided. This claim is not supported by the data.

5. Statistical reporting issues

Some t-values (e.g., t(14)=116.792) are mathematically implausible, suggesting errors in calculation or reporting. Multiple comparisons were performed without correction, inflating false positives.

Reviewer #2: Overall Assessment

This manuscript investigates an important clinical problem - anxiety during heroin withdrawal - and proposes GLP-1 signaling as a potential therapeutic target. The study is generally well-designed with appropriate behavioral tests and molecular analyses.

Sample Size and Statistical Power Concerns

The sample sizes are critically small and inconsistent across experiments. Western blot analyses use only n=3-4 for key proteins (GLP-1R, GLP-1, CREB), providing insufficient statistical power and vulnerability to outlier effects. Behavioral tests employ n=6-8, which is minimally adequate. The variation in sample sizes across figures without clear justification raises questions about animal exclusions. The authors should increase Western blot sample sizes to n=6-8 minimum or acknowledge this limitation explicitly.

Insufficient Verification of Viral Circuit Tracing

The AAV-Cre virus injection demonstrating NTS→BLA connectivity lacks critical validation. Missing elements include: stereotaxic atlas specification, injection accuracy quantification, verification of stable virus expression during testing, and quantification of what percentage of BLA-projecting NTS neurons are GLP-1-positive. Without these controls, it’s unclear whether observed effects truly result from this specific circuit or from non-specific viral effects and neighboring region involvement.

Inconsistent Drug Administration and Inadequate Pharmacological Controls

Drug administration protocols contain multiple problems. Heroin route is inconsistently reported (subcutaneous in Methods vs. intraperitoneal in abstract). Exendin-4 treatment uses only one dose without dose-response curves, the 7-day twice-daily schedule lacks pharmacokinetic justification, and persistence of effects after treatment cessation wasn’t tested. The “natural withdrawal” model isn’t verified beyond behavioral symptoms, and no naloxone-precipitated withdrawal control is included.

Specific Scientific Questions

1. Mechanism: How exactly does increased GLP-1 signaling cause anxiety? The authors suggest it’s through c-Fos activation, but the mechanism linking these is unclear.

2. Clinical relevance: The discussion mentions GLP-1 analogs causing anxiety in some patients (ref 27) but doesn’t adequately address how this affects the translational potential.

3. Alternative explanations: Could the anxiolytic effect of Exendin-4 in withdrawal mice simply be due to amelioration of withdrawal symptoms rather than a specific effect on anxiety circuits?

**Do you want your identity to be public for this peer review?** For information about this choice, including consent withdrawal, please see our Privacy Policy

Reviewer #1: **Yes:** MD MAJEDUR RAHMAN

Reviewer #2: No

---

## [Author Response · Author response to Decision Letter 1]

10 Dec 2025

Dear Editors and Reviewers:

Thank you for your letter and for the reviewers’ comments concerning our manuscript entitled “Exendin-4 enhances GLP-1 signaling and improves anxiety behavior in heroin withdrawal mice” (PONE-D-25-47998). Those comments are all valuable and very helpful for revising and improving our paper, as well as the important guiding significance to our researches. We tried our best to study comments carefully and have made some changes to the manuscript that we hope meet with approval. The changes have indicated by using red font colour in the paper.

Title: Exendin-4 enhances GLP-1 signaling and improves anxiety behavior in heroin withdrawal mice

Authors :Yang Xiang , Xiaowei Yan , Rongrong Li , Chunlu Li, Dan Yin , Cancan Luo, Yingqi Xiong, Yan Hong ,Yixin Li & Baijuan Xia

Point-by-point response to the reviewer’s comments:

Reviewer #1：

1. Conceptual contradiction

The introduction claims heroin dependence reduces glutamatergic neurons “without affecting anxiety or depression,” but the results later attribute increased anxiety during withdrawal to this reduction. This is inconsistent and requires clarification.

Response: We sincerely appreciate the valuable suggestions from the reviewers. According to existing research [Reference：doi:10.1038/s41572-019-0137-5; doi: 10.1016/j.b iopsych. 2022.04.007], when heroin is administered to patients, it significantly reduces negative emotions such as anxiety and depression in patients undergoing withdrawal. This is one of the main motivations for maintaining heroin dependence. When heroin-dependent patients enter the withdrawal period again, they will show significant anxiety and depression. Therefore, in combination with our research results, we believe that this is because the drug reward effect caused by heroin administration suppresses the anxiety and depression caused by the reduction of neurons. When the reward effect of heroin disappears, the anxiety and depression caused by the reduction of neurons will reappear. We may not have expressed this point clearly enough in the paper and have made supplements. Thank you to the reviewers for their careful reading.

2.Sex bias in study design

Only male mice were used. Given well-documented sex differences in addiction and anxiety behaviors, this omission limits translational value.

Response: We sincerely thank the reviewer for their careful reading. Due to the gender differences in behavioral tests between male and female mice (References: doi:10.1016/j.bbi.2017.02.006; doi:10.1038/s41583-021-00513-0), to avoid increasing the intra-group dispersion.In this work, we used male mice mainly to reduce biological variability related to the estrous cycle, we chose male mice of the same gender and stronger physical condition. This way, the statistical results obtained are more significant. We also acknowledge that using only male mice limits its translational value. In the future, we will strive to design a more comprehensive experimental plan.

3.Misinterpretation of depression-like behaviors

Reduced immobility in forced swim and tail suspension tests is interpreted as “reduced depression-like behavior,” but this could also reflect withdrawal-induced hyperactivity or agitation. The interpretation is misleading.

Response: We sincerely appreciate the reviewer’s thoughtful comment. Traditionally, increased immobility in the forced swim test and tail suspension test has been widely used as an index of despair-like or depression-like behavior in rodents (Reference：doi: 10.1523/JNEUROCSI.1767-22.2022; DOI: 10.1016/j.bbi. 2020.11.007).. While these assays remain classic and reliable tools, we fully agree that immobility can also be influenced by factors unrelated to depressive-like states, including discomfort or altered arousal during heroin withdrawal. Although our open-field results did not indicate increased locomotor activity, we acknowledge that the contribution of withdrawal-related agitation cannot be completely excluded. In future studies, we plan to include complementary behavioral measures, such as sucrose preference or novelty-suppressed feeding, to more comprehensively assess depressive-like behavior.

4.Mechanistic overstatement

The discussion repeatedly states Exendin-4 “restores glutamatergic neurons” and provides “neural repair,” yet no direct evidence of neurogenesis or synaptic remodeling was provided. This claim is not supported by the data.

Response: We sincerely thank the reviewer for their thorough review.We found that Exendin-4 significantly increased the number of glutamatergic neurons in the BLA brain area of heroin withdrawal group by counting and analyzing the GLP-1R mainly expressed glutamatergic neurons in each group of mice. We believe this is effective evidence and refer to the published literature (Reference：DOI: 10.3389/fnins.2022.70925; DOI: 10.1016/j.neubiorev.2022.104896), which reported that GLP-1 and its analog Exendin-4 have the effect of promoting neurogenesis and increasing synapses. Therefore, we did not continue to repeat this type of experiment and have revised the paper to address the situation mentioned. The changes in neural synapses are often associated with depressive emotions rather than anxiety emotions (Reference：doi: 10.1038/mp.2017.255; doi: 10.1038/nm. 4050), so we did not report it.We agree that direct evidence of neurogenesis or synaptic remodeling, e.g., PSD-95 quantification, electrophysiology et al. would strengthen the mechanistic interpretation, and we plan to address this more comprehensively in future studies. We sincerely thank the reviewer for pointing out this important issue.

5.Statistical reporting issues

Some t-values (e.g., t(14)=116.792) are mathematically implausible, suggesting errors in calculation or reporting. Multiple comparisons were performed without correction, inflating false positives.

Response: We thank the reviewer for pointing this out. Upon re-examination, we found errors in the calculation of some t-values and have corrected them. In addition, we have applied appropriate corrections for multiple comparisons to control the false-positive rate. The revised statistical results have been updated in the manuscript.

Reviewer #2：

1、Overall Assessment

This manuscript investigates an important clinical problem - anxiety during heroin withdrawal - and proposes GLP-1 signaling as a potential therapeutic target. The study is generally well-designed with appropriate behavioral tests and molecular analyses.

Sample Size and Statistical Power Concerns

The sample sizes are critically small and inconsistent across experiments. Western blot analyses use only n=3-4 for key proteins (GLP-1R, GLP-1, CREB), providing insufficient statistical power and vulnerability to outlier effects. Behavioral tests employ n=6-8, which is minimally adequate. The variation in sample sizes across figures without clear justification raises questions about animal exclusions. The authors should increase Western blot sample sizes to n=6-8 minimum or acknowledge this limitation explicitly.

Insufficient Verification of Viral Circuit Tracing

The AAV-Cre virus injection demonstrating NTS→BLA connectivity lacks critical validation. Missing elements include: stereotaxic atlas specification, injection accuracy quantification, verification of stable virus expression during testing, and quantification of what percentage of BLA-projecting NTS neurons are GLP-1-positive. Without these controls, it’s unclear whether observed effects truly result from this specific circuit or from non-specific viral effects and neighboring region involvement.

Inconsistent Drug Administration and Inadequate Pharmacological Controls

Drug administration protocols contain multiple problems. Heroin route is inconsistently reported (subcutaneous in Methods vs. intraperitoneal in abstract). Exendin-4 treatment uses only one dose without dose-response curves, the 7-day twice-daily schedule lacks pharmacokinetic justification, and persistence of effects after treatment cessation wasn’t tested. The “natural withdrawal” model isn’t verified beyond behavioral symptoms, and no naloxone-precipitated withdrawal control is included.

Response: We sincerely appreciate your thorough review.

1.Sample Size and Statistical Power Concerns

We thank the reviewer for the comment. We have explicitly acknowledged the limitation of small sample sizes for Western blot analyses in the limitations section of the manuscript. For the apparent variation animal numbers across experiments, we have supplemented the experimental sample size in Figure 2. In Figure 5, each group used n=6 for Exendin-4 dose exploration to minimize animal use in accordance with ethical considerations; we have clarified this rationale in the Methods section.

2. Insufficient Verification of Viral Circuit Tracing

We thank the reviewer for this helpful comment.We acknowledge that the evidence in the virus tracking circuit needs further optimization. In the larger image, we only provided a partial view of the outer side of the substrate for the stereotaxic map, as displaying the entire cross-sectional view is difficult to see the localization situation clearly or requires more space. Therefore, we chose to fold it in; We used multiple mice for stereotactic injection in the early stage, and compared the maps to ensure accurate injection positions before conducting virus injection, so we did not quantify the accuracy of injection. Our research in this area is relatively shallow and did not take into account the validation of stable virus expression during testing, as well as the quantification of the percentage of GLP-1 positivity in NTS neurons projected to BLA. However, we conducted immunofluorescence staining of GLP-1 R in BLA brain regions, WB detection of GLP-1 projected to BLA, and found reliable references as references (Reference：doi: 10.1111/bph. 15682; doi：10.1186/s13578-00914-3) to demonstrate the existence of a pathway between NTS-BLA, thus proving the reliability of our experimental conclusions in this part.

3. Inconsistent Drug Administration and Inadequate Pharmacological Controls

We thank the reviewer for pointing this out. We have corrected the heroin administration route in the abstract—it should be subcutaneous, not intraperitoneal. The Exendin-4 dose and schedule (twice daily for 7 days) were chosen based on blood glucose and body weight monitoring, as well as published studies (DOI: 10.3389/fnsys. 2021.711750; DOI: 10.3390/nu15038; DOI: 10.1016/j.psyneuen.2015.11.021). Through behavioral testing, we found that the anxiety like behavior of heroin withdrawal mice improved before determining this experimental plan. We acknowledge that we only used a single dose and did not provide dose-response curves or pharmacokinetic support, which limits the credibility and promotional value of our protocol. The reason why the natural withdrawal model was only validated at the behavioral symptom level and did not include a control measure for naloxone induced withdrawal is that our modeling protocol referred to published literature reports and had previously conducted relevant experiments in the laboratory to test the feasibility of the model using naloxone (Reference：DOI: 10.1016/j. aneulet. 2022.137934), so we did not include a control measure for naloxone induced withdrawal.We will consider more rigorous pharmacological validation in future studies.

2、Specific Scientific Questions

(1). Mechanism: How exactly does increased GLP-1 signaling cause anxiety? The authors suggest it’s through c-Fos activation, but the mechanism linking these is unclear.

Response: We thank the reviewer for the valuable feedback. According to the published literature and pathway databases （ https://www.kegg.jp/kegg/pathway.html ） It is believed that GLP-1 can act on its receptor GLP-1 R to regulate the expression of BDNF, TrkB, CREB. These signaling events can then promote c-Fos expression , leading to neuronal activation and contributing to the regulation of anxiety-like behaviors (DOI: 10.1523/JNEUROSCI. 1767-22.2022；DOI 10.3389/fendo.2023.1268865). While this provides a plausible pathway, we acknowledge that the precise mechanistic link between GLP-1 signaling and anxiety requires further experimental validation, which we will explore in future studies

(2). Clinical relevance: The discussion mentions GLP-1 analogs causing anxiety in some patients (ref 27) but doesn’t adequately address how this affects the translational potential.

Response: We thank the reviewer for careful reading. We have discussed this point in the limitations section of the manuscript.While Exendin-4 improved anxiety-like behavior in our mouse model, clinical reports suggest that GLP-1 analogs can sometimes induce or worsen anxiety in patients (ref. 27). This implies that, in future translational applications, careful attention is needed: patients may experience increased anxiety during treatment, which could reduce medication tolerance or even lead to depression. Therefore, dosing, monitoring, and patient selection should be carefully considered to minimize potential adverse effects.

(3) . Alternative explanations: Could the anxiolytic effect of Exendin-4 in withdrawal mice simply be due to amelioration of withdrawal symptoms rather than a specific effect on anxiety circuits?

Response: Thank you for the valuable feedback from the reviewer. We acknowledge that our experimental design cannot fully distinguish whether the anxiolytic effect of Exendin-4 is due to direct modulation of anxiety circuits or secondary to alleviation of general withdrawal symptoms,as the drug was administered systemically rather than via localized brain delivery. In existing research reports, we have not seen any reports on the use of Exendin-4 to improve heroin withdrawal symptoms. However, based on the pharmacological mechanism of Exendin-4, there is a possibility of this. We have acknowledged the shortcomings in this regard in the limitations section of the paper and will plan to use more targeted experimental approaches in future studies to clarify the mechanism.

---

## [Editor Report · Decision Letter 1]

16 Feb 2026

Exendin-4 enhances GLP-1 signaling and reduces anxiety-like behaviors in male heroin withdrawal mice

PONE-D-25-47998R1

Dear Dr. Xia,

We’re pleased to inform you that your manuscript has been judged scientifically suitable for publication and will be formally accepted for publication once it meets all outstanding technical requirements.

Kind regards,

Hira Rafi

Academic Editor

PLOS One
---

## [Editor Report · Acceptance letter]

PONE-D-25-47998R1

PLOS One

Dear Dr. Xia,

I'm pleased to inform you that your manuscript has been deemed suitable for publication in PLOS One. Congratulations! Your manuscript is now being handed over to our production team.

Kind regards,

on behalf of

Dr. Hira Rafi

Academic Editor

PLOS One